# SAM meets Gaze: Passive Eye Tracking for Prompt-based Instance Segmentation

**Daniel Beckmann**[*1,2]                      DANIEL.BECKMANN@UNI-MUENSTER.DE  and
**Jacqueline Kockwelp**[*1,2,3]                      J.KOCKWELP@UNI-MUENSTER.DE
**Joerg Gromoll**[3]                      JOERG.GROMOLL@UKMUENSTER.DE
**Friedemann Kiefer**[4]                      FKIEFER@UNI-MUENSTER.DE
**Benjamin Risse**[1,2]                      B.RISSE@UNI-MUENSTER.DE
[1]*Institute for Geoinformatics, University of Münster*

[2]*Faculty of Mathematics and Computer Science, University of Münster*

[3]*Centre of Reproductive Medicine and Andrology, University Hospital Münster*

[4]*European Institute for Molecular Imaging, University of Münster*

## Abstract

The annotation of large new datasets for machine learning is a very time-consuming and expensive process. This is particularly true for pixel-accurate labelling of e.g. segmentation masks. Prompt-based methods have been developed to accelerate this label generation process by allowing the model to incorporate additional clues from other sources such as humans. The recently published Segment Anything foundation model (SAM) extends this approach by providing a flexible framework with a model that was trained on more than 1 billion segmentation masks, while also being able to exploit explicit user input. In this paper, we explore the usage of a passive eye tracking system to collect gaze data during unconstrained image inspections which we integrate as a novel prompt input for SAM. We evaluated our method on the original SAM model and finetuned the prompt encoder and mask decoder for different gaze-based inputs, namely fixation points, blurred gaze maps and multiple heatmap variants. Our results indicate that the acquisition of gaze data is faster than other prompt-based approaches while the segmentation performance stays comparable to the state-of-the-art performance of SAM. Code is available at `https://zivgitlab.uni-muenster.de/cvmls/sam_meets_gaze`.

**Keywords:** Instance Segmentation, Image Annotations, Gaze Prompts, Segment Anything Model (SAM), Deep Learning, Eye Tracking

## 1. Introduction

Being one of the most popular tasks in computer vision today, image segmentation has experienced huge improvements over the years Dong et al. (2021); LeCun et al. (2015). Recently a new foundation model for prompt-based segmentation tasks called "Segment Anything Model" (SAM) was introduced by  Kirillov et al. (2023). Developed to segment objects in a class-agnostic fashion, training the model required huge amounts of annotated data. Together with the model, the authors provided an efficient interactive data collection loop using the foundation model, resulting in a new dataset of 11 million images with a total of 1 billion segmentation masks. However, user input is still required for SAM to identify an object of interest, which subsequently gets segmented. For SAM, this input is accepted as

---

* contributed equally and ordered alphabetically

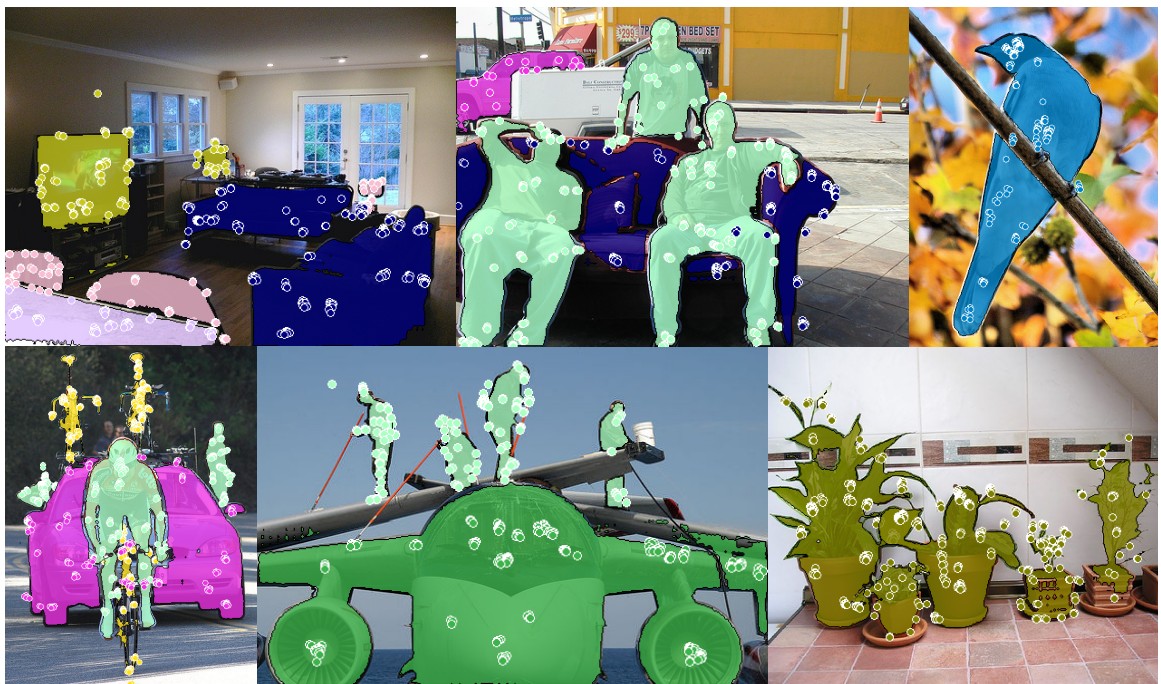

Figure 1: Example results for SAM meets Gaze using fixation-based prompts.

either point coordinates, a bounding box or a rough mask of the object to be segmented, or a combination of the aforementioned. This type of interactive prompting has been studied for a wide range of interaction schemes and prompt types, such as bounding boxes Kulharia et al. (2020); Dai et al. (2015); Khoreva et al. (2017); Tian et al. (2021), scribbles Lin et al. (2016); Peng et al. (2020) or points Papadopoulos et al. (2017a); Bearman et al. (2016); Xu et al. (2016); Maninis et al. (2017); Koohbanani et al. (2020). Depending on the object's fidelity and input type, providing even these sparse annotations can be time consuming and cumbersome. In addition, in the case of domain-specific image data, such as medical images, experts in the field are required to explicitly provide the annotations, further increasing the overall annotation cost Chartrand et al. (2017). Since the first and foremost interaction during all image annotations is the visual inspection of the images, gaze-based interaction appears to be a compelling candidate for prompt-based annotation, especially since passive eye tracking systems do not require active physical interaction with any additional hardware. Therefore, the extraction of gaze prompts are naturally integratable into various image analysis workflows while providing an efficiency beyond all other explicit data interaction strategies.

## 1.1. Contribution

In this paper, we explore the use of eye tracking information as a prompt for SAM. We integrate a passive screen-based eye tracking system into the workflow, which allows the human annotator to quickly and conveniently provide gaze trajectories as a prompt for segmenting

a specific instance (see Figure 1). Following the approach of Kockwelp et al. (2023), no specific task or time limit was given to the human annotator so that an unrestricted inspection of the object of interest was possible. For seamless integration we convert the passively captured raw gaze data into blurred gaze maps, heatmaps and extracted fixations and fine-tuned the sparse and dense prompt encoder of SAM with each prompt type, respectively. We use the existing gaze dataset by Kockwelp et al. (2023) for training and evaluation and complement it by also annotating the PascalVOC2012 validation Everingham et al. (2012) dataset resulting in a total of $7,434$ instance annotations. Our analysis of the annotation process and the results of the segmentation performance show that using gaze data as a prompt for SAM can be beneficial for image segmentation by being less time consuming while reaching comparable performances to the other prompt-based options. We therefore conclude that the usage of eye tracking data as a prompt for SAM is a convenient alternative input strategy for SAM given its potential to gather additional user-input without requiring any conscious interaction in a variety of critical applications.

## 1.2. Related Work

This work mainly touches two subtopics of image analysis, namely the utilization of gaze data in supervised machine learning methods and, more generally, the field of prompt-assisted segmentation methods.

**Gaze Data in Supervised Machine Learning** The integration of gaze data into image analysis algorithms has a relatively long history Mathe and Sminchisescu (2014). In the field of image classification, eye tracking data can be used as auxiliary information to improve model performance. For example, Saab et al. (2021) used gaze data to improve medical image classifications and to estimate weak classification labels based on statistically derived gaze features. Similarly, Wang et al. (2017) integrated gaze data to guide the region selection towards local information in a weakly supervised image classification framework. Karessli et al. (2016) also used gaze data implicitly by converting the raw gaze points into histograms and grid-based features which are subsequently used for zero-shot classification. To incorporate expert knowledge, Stember et al. (2020) combined gaze data with speech recognition to create point annotations for brain lesion localization in MRI scans, which later were used to train a neural network on point location prediction. Murrugarra-Llerena and Kovashka (2017) showed that gaze data can be used to predict visual attributes of objects, such as "open" or "pointy". With the introduction of modern transformer models by Vaswani et al. (2017), attention-based methods gained popularity in all kinds of machine learning tasks. Interpreting gaze data as human attention, eye tracking information has been used to guide the attention of the model used in medical computer-assisted diagnosis Wang et al. (2022). Specifically, a consistency module has been used to regularize the attention of the model during training based on the provided gaze data from experts. Similarly, Bhattacharya et al. (2022) trained a novel student-teacher architecture with an explicit visual attention loss built upon attention derived from human gaze. For video captioning, Yu et al. (2017) incorporated a gaze encoding network into their model which generates spatial and temporal attention patterns based on the provided gaze information.

**Prompt-assisted Segmentation** Improving segmentation models by leveraging additional user input has been subject of many investigations. Together with bounding boxes

and scribbles, point-based prompts are the most popular and widely studied input types for segmentation models and are usually provided to select individual target objects. Bounding boxes can be used as a noisy label to improve boundary refinement Kulharia et al. (2020) or to constrain a segmentation using a euclidean distance transformation based on the given bounding box as an additional input channel to the neural network Xu et al. (2017). Similarly, Peng et al. (2020) showed that scribble annotations can be used to initialize a contour deformation process to obtain object masks. Several works have studied the use of point-based auxiliary input. For example, Koohbanani et al. (2020) utilized clicks or scribbles to perform interactive segmentation for medical images and Bearman et al. (2016) combined point supervision with an objectness prior for semantic segmentation. While Bearman et al. allowed points anywhere on the object, Papadopoulos et al. (2017b) analyzed the usage of extreme points for segmentation tasks. In particular, the extreme points were used to initialize a GrabCut-like segmentation algorithm. This idea got extended by Maninis et al. (2017), who integrated extreme points into a machine learning-based segmentation model called DEXTR. The recent Segment Anything model allows both bounding boxes and arbitrary points as input. Furthermore, it can be guided by providing the prediction from a previous iteration as an additional mask input, when used in an iterative fashion Kirillov et al. (2023). In parallel to our work, Wang et al. investigated the use of gaze data for SAM by comparing the zero-shot capabilities of SAM using gaze-based point prompts for interactive data annotation and classical mouse-based interaction Wang et al. (2023).

## 2. Materials and Methods

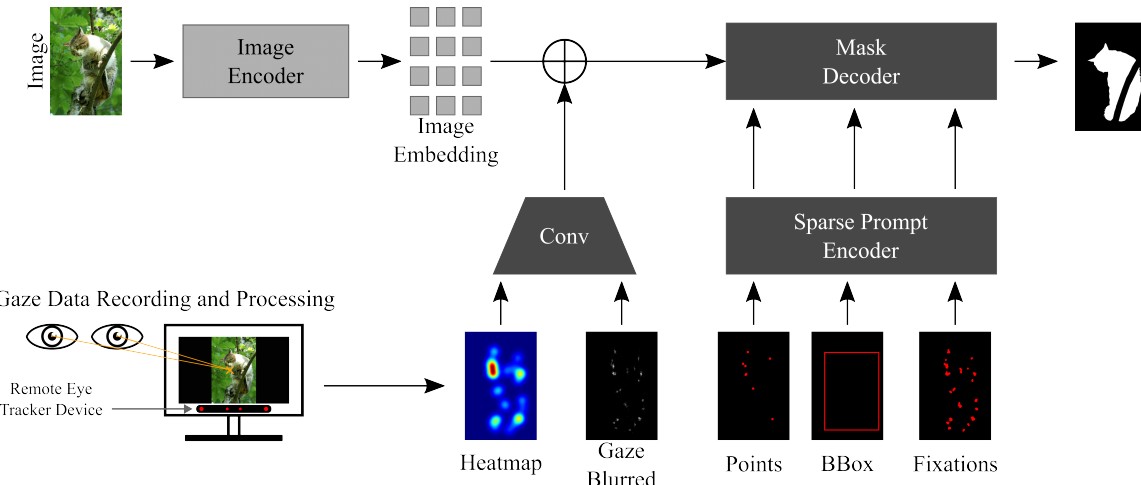

Figure 2: Method Overview. An image embedding is calculated using the original heavy-weight image encoder of SAM. Additionally, new eye tracking based prompt options are generated using a remote eye tracking device ("Gaze Blurred", fixation points and Heatmaps). The sparse and dense prompt encoder (Conv) and the mask decoder are finetuned for each new prompt.

An overview of our method is given in Figure 2. The general idea is to finetune the prompt encoders and the mask decoder of original SAM for different gaze-based prompts. Particularly, we investigate two variants of gaze inputs to the model, namely mask- and point-based prompts. For the mask-based version, we test four different types of input masks, which are constructed directly from the recorded gaze. The first mask, in the following referred to as "Blurred Gaze", is obtained by first constructing a binary mask from all gaze points by setting all pixels containing a recorded gaze point to 1 (independent from the number of gaze points over time on the pixels).

Subsequently, the binary masks are slightly blurred using a Gaussian kernel of radius 3 and $\sigma = 5$, similar to the blurring of extreme points used in Maninis et al. (2017). For the other three mask types, we follow the heatmap construction described in Section 2.3, using the raw gaze points ("Heatmap Gaze"), the extracted fixation points ("Heatmap Fixations") and the fixations with their corresponding duration ("Heatmap Fixations + Duration") as input for the heatmap generation. Finally, following Kirillov et al. (2023), masks are downsampled by a factor of four and used as a dense input to the prompt encoder.

For point-based prompts, we use the calculated fixation point coordinates (see Section 2.2) as a sparse input. In contrast to the original SAM, we only provide foreground points, setting the labels for the prompt encoder accordingly. Using the raw gaze point coordinates instead of fixations as a sparse input results in very high memory consumption due to a comparatively high number of gaze points, which makes training inefficient (see Section 2.4). Therefore, we restrict our experiments with point-based inputs to fixations only.

All experiments use the publicly available pretrained ViT/H variant of SAM as a baseline, unless stated otherwise. To reduce computational cost and training time, we freeze the heavyweight image encoder of SAM, training only the lightweight downstream modules. This allows us to precompute the embeddings for all images, enabling a significantly increased batch size during training due to the substantially smaller memory footprint and speeding up the training procedure by a factor of more than ten. To study the suitability of different gaze prompts for SAM, we intentionally use a non-interactive training procedure (i.e. no iterative procedure or refinement during each iteration is used). We compare our gaze-based prompts to finetuned versions of SAM using either bounding boxes or sampled foreground points as input. For bounding boxes, we use the ground truth box for prompting, with no additional noise. For point-based input, we randomly sample eight points from the ground truth mask and use them as input.

## 2.1. Dataset

For training and evaluation of our method we use the following three datasets: PascalVOC2012 train, PascalVOC2012 validation Everingham et al. (2012) and 500 cell objects of the Cellpose dataset Stringer et al. (2021). Gaze annotations for PascalVOC2012 train and Cellpose are taken from an existing dataset Kockwelp et al. (2023). We extend these sets and annotate the PascalVOC2012 validation set with ten human annotators (8 self-identified as male, 2 as female, mean age $\mu = 28.0$, $\sigma = 1.82$) and increase the amount of data by $1,449$ images and $3,427$ instance annotations. Each human annotator has been

informed about the use of their recorded data and has given their written consent for the processing and publication of the data.

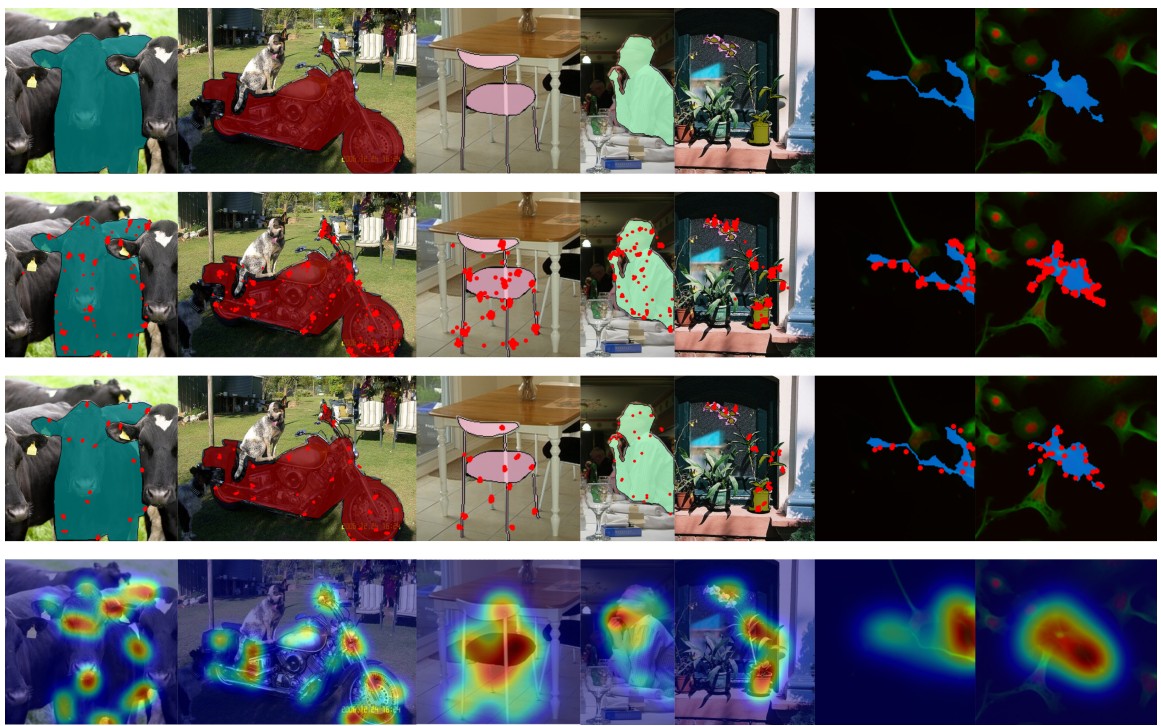

Figure 3: Dataset examples for PascalVOC2012 validation and Cellpose. **First row:** Images with ground truth mask. **Second row:** Images with raw gaze point map visualization. **Third row:** Images with fixation point map visualization. **Fourth row:** Images with heatmap visualization.

To ensure full comparability, these additional gaze trajectories and annotations are generated in the same way as described in Kockwelp et al. (2023), namely by using a screen-based Tobii Pro Fusion eye tracking system that passively records gaze data with a sampling frequency of 120 Hz. Before a human annotator could start the annotation process, the eye tracker was recalibrated specifically for the user. For the calibration, five points (centre and four corners of the screen) were displayed on the screen one after the other and the human annotator was asked to look at a point until it disappeared. After all points were displayed and a calibration was calculated and applied to the device, a separate validation was started in which the human annotator was again asked to look at five points on the screen one after the other. The whole process including the validation takes about 25 seconds. A recalibration for the same user was performed for each annotation session, e.g. to adapt to changing lighting conditions. Before every single annotation a bounding box and the ground truth mask was shown for 0.5 seconds to guide the user to the current object of interest and the user was given the instruction to inspect the entire object as required to familiarize himself/herself with the covered area. We note that these inspections are

also necessary before starting conventional annotation purposes and we do not restrict the inspection process to any further instruction or time limits. Instead, the human annotator is allowed to freely start and end the gaze recording by clicking a keyboard key. In case the annotator is distracted or it is unclear which object should be inspected, the annotation can be repeated. For very small objects like bottles we initially zoom into the image.

Figure 3 (first row) shows some example images of the PascalVOC2012 validation and the Cellpose dataset with overlayed ground truth masks. The second row shows a visualization of all recorded raw gaze points. It can be seen that the whole object of interest is scanned by the human annotators and that the gaze data covers the entire object including its extremities. Using an I-VT filter (see Section 2.2), we reduce the noise level and identified the fixation points which are visualized in the third row. Additionally, we transform the raw gaze points as well as the calculated fixation points into different heatmaps following the description in Section 2.3. Colour-coded heatmap examples are given in the fourth row.

## 2.2. Fixations

To identify fixations, we use the Velocity-Threshold Identification (I-VT) filter which identifies saccades and fixation points based on their point-to-point velocities Salvucci and Goldberg (2000). Saccades normally occur very fast with a very high velocity, whereas fixations are more aggregated with a low velocity. In a first step, we calculate the point-to-point velocities for each recorded gaze point using the next or previous point. Afterwards, we apply a velocity threshold to classify each gaze point as a saccade when its velocity is above an experimentally selected threshold, and otherwise as a fixation. In a final step all identified consecutive fixation points are aggregated to a single fixation with the representation $(x, y, t, d)$, where $x, y$ is the centroid of the grouped fixation points, $t$ the starting time and $d$ the duration of the fixation. All identified saccade points are removed to reduce the noise level.

## 2.3. Gaze Heatmaps

Heatmaps are a type of visualization often used for the analysis of eye tracking data. Generally, it is a colour-coded mask that is generated depending on the frequency and duration of viewing certain regions. This allows a quick and intuitive capture and analysis of the most important areas of an image that have received particular attention. For this work, we generate three different heatmaps: raw gaze point based ("Heatmap Gaze"), fixation based ("Heatmap Fixations") and fixation based using their corresponding duration ("Heatmap Fixations + Duration") . The basic procedure is the same for all of them. First we calculate a Gaussian kernel (not normalised) with a width and height of 100 pixels and a standard deviation of $\sigma_{HM} = \frac{100}{6}$, which is then added to the heatmap at the corresponding position for each raw gaze or fixation point. This makes areas where many raw gaze points occur much more important than surrounding areas due to the accumulation of signal from overlapping kernels. To have the same effect for the condensed fixation points, we multiply the Gaussian kernel with the fixation duration before adding it to the heatmap for the corresponding fixation location.

## 2.4. Training Details

For all experiments on PascalVOC2012, we train the models for 75 epochs. For experiments with the smaller Cellpose 500 dataset, the models are trained for only 55 epochs. When using new gaze-based prompts we use a learning rate of $8 \cdot 10^{-4}$, for models trained with established prompts we lower the learning rate to $2 \cdot 10^{-4}$. More details on the impact of different learning rates can be seen in Figure 6. In all experiments, an initial linear warm-up of 30 steps is used, followed by an exponential learning rate decay of 0.95 after every epoch. We use the proposed loss configuration from SAM, i.e.

$$\mathcal{L} = \mathcal{L}_{IOU} + \mathcal{L}_{Dice} + 20 \cdot \mathcal{L}_{Focal}. \tag{1}$$

For our main finetuning results with gaze-based prompts on PascalVOC2012 and Cellpose 500, we use two A100 (40GB) graphics cards for training and a batch size of 12 per GPU. The batch size is equal for all input modalities to ensure comparable results. Due to the varying number of fixations per object, the input tensors containing the point coordinates have to be padded for batch training, causing a bigger memory footprint during training than would be necessary during inference when using fixations as one of the prompts. Using all available raw gaze points instead of fixations as a prompt input results in a batch size of 1 sample per GPU, rendering training extremely inefficient. The reserved label $-1$ is assigned to the artificial points added to equalize the size of the tensors, where 0 indicates the background and 1 indicates the foreground in the normal case. After calculating the positional embedding of each point, the embeddings for points with label $-1$ are substituted with a learned "no point" embedding. This results in the fixation-based prompt being the most resource-demanding training modality. For inference, when done separately for each object, the padding of point inputs is no longer required.

## 3. Results

### 3.1. Annotation Efficiency

The average per-image annotation time for all three datasets is 5.68 seconds (PascalVOC2012 train: 6.33, PascalVOC2012 val: 5.59, Cellpose: 5.13). This is much faster than drawing a polygon mask or a bounding box which takes an average of 79 seconds or 35 seconds respectively Papadopoulos et al. (2017b). Gaze annotaions are also about about two seconds faster than using four clicks to provide extreme points of the object of interest which takes about 7.5 seconds Maninis et al. (2017). The annotation time also seems to depend heavily on the human annotator, as the mean time for the 11 annotators ranges from 2.86 to 7.79 seconds. There are also major annotation time differences between the individual classes of the three datasets. For example, it takes the least time to annotate an object of the class bottle with an average of 4.23 seconds and the most time to annotate an object of the class horse with an average of 8.67 seconds. A more detailed analysis of the annotation time per human annotator and class category can be found in appendix Figure A.1 and Figure A.2.

In order to analyse the class-related differences in more detail, we examine the individual classes with regards to their convexity and area size (see Figure 4). The convexity $c$ of an object is defined as follows:

$$c = \frac{\text{area of gt mask}}{\text{convex hull area of gt mask}}, \ c \in [0, 1]. \tag{2}$$

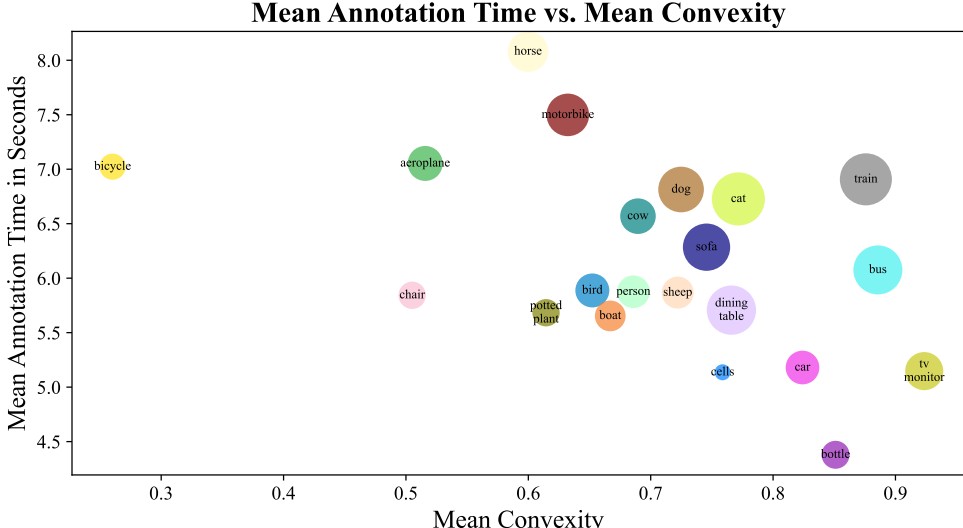

Figure 4: Mean annotation time vs. mean convexity grouped by class category. The circle sizes represent the mean area for every class.

There are large differences between the convexity of the individual classes, which clearly have an effect on the required annotation time. Very complex objects with finer structures such as bicycles require significantly more time than less granular objects such as buses, trains, tv monitors and bottles. The area size of an object, which is illustrated with the size of each point marker in Figure 4, also seems to have a huge influence on the inspection time of an annotator. Very small objects such as cells or bottles take relatively little time while larger objects, although highly convex, take much more time to inspect.

It should be noted that the increased annotation times for more complex object geometries are of course also required when using other annotation strategies such as drawing polygons. Moreover, the visible inspection of the object is a prerequisite for all image labelling strategies. Therefore, if integrated into an implicit image inspection routine, gaze prompts would not require any additional annotation time.

## 3.2. Gaze Accuracy

In contrast to conventional prompt-based methods such as bounding boxes, single points or scibbling, which are often generated via computer mouse or tablet input, eye tracking is usually less accurate. The mean precision for the validation of the calculated calibrations of all human annotators is $0.23°$ (median: $0.21°$, standard deviation: $0.13°$) and the mean accuracy is $0.40°$ (median: $0.38°$, standard deviation: $0.19°$). We analyse the gaze accuracy in more detail for the datasets used in this work with regards to the individual human annotators and the individual class categories.

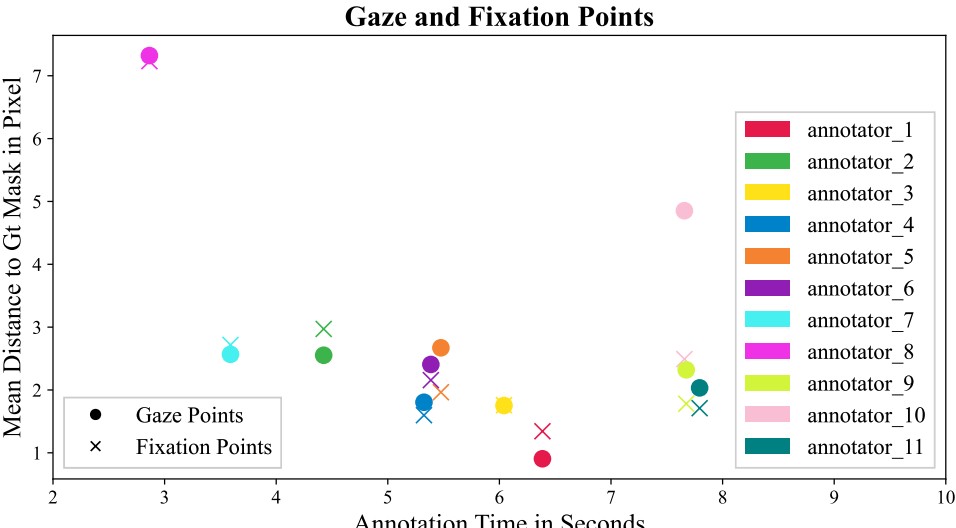

Figure 5: Mean distance between gaze data and ground truth mask grouped by human annotators.

Figure 5 shows the mean distance of the individual gaze and fixation points to the ground truth mask depending on the mean inspection time for every annotator. It is noticeable that for the majority of the gaze points the deviation is very small. Annotator_8 is an outlier showing the lowest annotation time (2.8 seconds per object) while also having the highest inaccuracy. A more detailed analysis is shown in the appendix Figure A.3, which shows the accumulated frequency in percentage of gaze points (see Figure A.3($a$)) and fixations (see Figure A.3($b$)) grouped by their distance to the ground truth mask. The same analysis per class category can be found in appendix Figure A.4. It should be noted that in the PascalVOC2012 dataset there is a five pixel wide boundary around each object, which is declared as void pixels, meaning these pixels may or may not belong to the object. Furthermore, there are no major differences between the gaze and fixation points in terms of accuracy. This is due to the fact that the saccades, which happen very fast, consist of only a few gaze points so that they only account for a very small percentage of all gaze points.

### 3.3. Segmentation Performance

We evaluate gaze-based prompts on PascalVOC2012 and Cellpose 500 to investigate segmentation capability for both natural and medical images. Starting with natural images of everyday objects, the results shown in Table 1 for PascalVOC2012 demonstrate the effectiveness of gaze for prompting, with the performance of fixation-based prompts being close to points and bounding boxes as used in SAM. Furthermore, we achieve only slightly worse results when using mask-based prompts with little variance regard to different mask types.

Table 1: PascalVOC2012 Segmentation Results. Fixations achieve the best performance out of gaze-based prompt types. Mask-based prompt variants are very similar in performance, and combining fixations with the best mask prompt (last row) does not increase overall performance.

| Prompt Type | Zero-Shot | | Finetuned | |
| --- | --- | --- | --- | --- |
| | IOU | F1 (Dice) | IOU | F1 (Dice) |
| Points | $0.813_{\pm0.003}$ | $0.883_{\pm0.002}$ | $\mathbf{0.850}_{\pm0.001}$ | $\mathbf{0.909}_{\pm0.001}$ |
| BBox | $0.827_{\pm0.005}$ | $0.891_{\pm0.004}$ | $0.844_{\pm0.002}$ | $0.903_{\pm0.001}$ |
| Fixations | $0.677_{\pm0.009}$ | $0.778_{\pm0.008}$ | $\mathbf{0.825}_{\pm0.003}$ | $\mathbf{0.887}_{\pm0.002}$ |
| Blurred Gaze | 0 | 0 | $0.812_{\pm0.004}$ | $0.878_{\pm0.003}$ |
| HM Gaze | 0 | 0 | $0.807_{\pm0.008}$ | $0.873_{\pm0.008}$ |
| HM Fixations | 0 | 0 | $0.809_{\pm0.005}$ | $0.873_{\pm0.005}$ |
| HM Fix. + Dur. | 0 | 0 | $0.804_{\pm0.005}$ | $0.870_{\pm0.004}$ |
| Fixations + Blurred Gaze | $0.369_{\pm0.002}$ | $0.484_{\pm0.003}$ | $0.822_{\pm0.002}$ | $0.886_{\pm0.001}$ |

We also evaluate the proposed gaze prompts using the original SAM model. Fixation points share certain similarities to conventional point-based prompting and achieve an Intersection over Union (IOU) score of 0.677 even without any finetuning. Even though the number of fixations usually exceeds the eight points used in classical point prompting, the inherent inaccuracies of fixations (see Section 3.2) can lead to a performance drop. In its original form, SAM expects points to be labeled as fore- or background. This constraint is violated when using fixations, since all fixations are interpreted as being foreground, despite inaccuracies in the border region of the object. Regarding mask-based gaze prompts, the original SAM model fails to segment objects without finetuning. Originally, the dense mask encoder is used to iteratively refine the predicted mask, receiving the previous prediction as an input. In contrast, the gaze-based masks are either sparse (when using the blurred raw gaze point approach) or more inaccurate for the heatmap-based types. In both cases the confidence in predicted masks is low so that empty masks are returned after thresholding. Finally, we investigate combining the most promising mask-based prompt input (i.e. the blurred raw gaze point approach) with fixations. Interestingly, this multi-modal input does not improve the segmentation quality. The performance metrics are very similar to the fixation-only approach.

Analogously, fixations as prompts result in similar performance for Cellpose 500 in comparison to points and bounding boxes, albeit on a lower overall performance (see Table 2, left side). Since SAM was originally trained mostly with photographs of real world scenes, performance on new image modalities like medical microscopy images is lower. This gap is present regardless of the prompt type and more severe for models without finetuning. Despite the minimal size of the Cellpose 500 training dataset with 81 images and 402 objects only, good segmentation performance is achieved after finetuning for both classical and gaze-based prompts. In contrast to PascalVOC2012, the gap between mask-based

Table 2: Cellpose 500 Segmentation Results. For the two left columns, the original SAM model is used as a baseline. On the right, the model was pretrained using PascalVOC2012 and the corresponding gaze prompt. Fixations perform best, similar to PascalVOC2012. Training with blurred gaze masks as input was unstable, hence results are omitted. The other mask types achieve comparable results.

| | Zero-Shot | | Finetuned | | Zero-Shot[Pascal] | | Finetuned[Pascal] | |
|---|---|---|---|---|---|---|---|---|
| **Prompt Type** | IOU | F1 (Dice) | IOU | F1 (Dice) | IOU | F1 (Dice) | IOU | F1 (Dice) |
| Points | 0.369 | 0.506 | 0.719 | 0.833 | 0.322 | 0.463 | **0.713** | **0.829** |
| BBox | 0.480 | 0.632 | **0.728** | **0.837** | 0.398 | 0.543 | 0.707 | 0.823 |
| Fixations | 0.325 | 0.471 | **0.709** | **0.825** | 0.191 | 0.299 | **0.693** | **0.814** |
| Blurred Gaze | *no training* | | | | 0.179 | 0.284 | 0.604 | 0.750 |
| HM Gaze | 0 | 0 | 0.602 | 0.747 | 0.100 | 0.163 | 0.620 | 0.760 |
| HM Fixations | 0 | 0 | 0.576 | 0.724 | 0.100 | 0.162 | 0.606 | 0.749 |
| HM Fix. + Dur. | 0 | 0 | 0.584 | 0.732 | 0.080 | 0.126 | 0.612 | 0.755 |

and point-based gaze input is bigger and competitive performance is only achieved by using fixations as prompts. In addition, training with blurred raw gaze point masks is unstable and often diverges. We also investigate potential benefits of pretraining the model with gaze-based prompts on PascalVOC2012 as shown in Table 2 on the right. While the performance after finetuning is slightly lower for all input modalities, the segmentation quality in the zero-shot scenario can be increased for mask-type input prompts. Pretraining with PascalVOC2012 also enables the training for the blurred gaze mask prompt altogether. This indicates that content-agnostic pretraining using gaze prompts might be beneficial to establish the new prompt types before finetuning. Given the very limited Cellpose 500 dataset size, further investigation is required.

Table 3: Ablation Study on PascalVOC2012. Leaving out either prompt encoder or mask decoder degrades performance by a similar amount, with the exception of the blurred gaze mask, which heavily relies on finetuning the prompt encoder.

| **Prompt Type** | w/o Encoder | w/o Decoder | Full |
|---|---|---|---|
| Fixations | 0.888 | 0.875 | **0.887** |
| Blurred Gaze | 0.443 | 0.810 | **0.878** |
| HM Gaze | 0.832 | 0.823 | **0.873** |
| HM Fixations | 0.822 | 0.826 | **0.873** |
| HM Fix. + Dur. | 0.821 | 0.817 | **0.870** |

To analyse the impact of the prompt encoder and mask decoder on segmentation quality, we conduct a small ablation study, shown in Table 3, on PascalVOC2012 by freezing either the encoder or decoder, respectively. For all configurations, performance is inferior to the

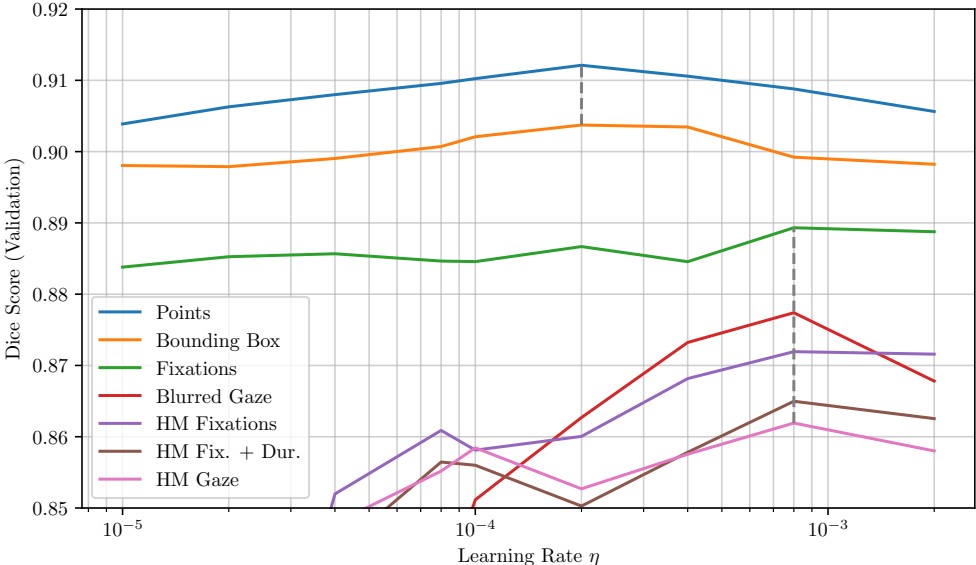

Figure 6: Influence of learning on segmentation performance. For original prompt types, a lower learning rate of $2 \cdot 10^{-4}$ is sufficient. For new prompt types, an initial learning rate of $8 \cdot 10^{-4}$ works best.

full model. Interestingly, leaving out either component leads to very similar performance drops, with the exception of blurred gaze masks, where training without the encoder fails to learn high-quality segmentations.

Considering different learning rates for the finetuning process, training results are quite stable. Nevertheless, as shown in Figure 6, the originally proposed learning rate of $8 \cdot 10^{-4}$ remains best for finetuning to new prompt types, with slight performance degradation for higher and lower learning rates. For mask-based prompts, learning rates below $10^{-4}$ result in significant performance drops. Intuitively, finetuning for the known prompt types requires a lower learning of $2 \cdot 10^{-4}$ for best results.

In comparison to EyeGuide Kockwelp et al. (2023), performance on PascalVOC2012 could be improved from an IOU of 0.764 to 0.825. It should be pointed out that a direct comparison might not be possible due to the difference in available training data and pre-training effort. EyeGuide was evaluated on 20% of the PascalVOC2012 train dataset using the remaining 80% for training while our results (reported in Table 1) refer to performance on the validation set, with the full training set available for training. In contrast, performance on Cellpose 500 is worse, with an IOU of 0.836 achieved by EyeGuide and 0.709 when using fixations as a prompt for our model. In general, even with the original prompting methods, SAM fails to achieve performance similar to EyeGuide on Cellpose 500. We hypothesize that this is due to the fact that the heavyweight image encoder of SAM was

trained using natural images almost exclusively, resulting in a performance bias towards this image modality.

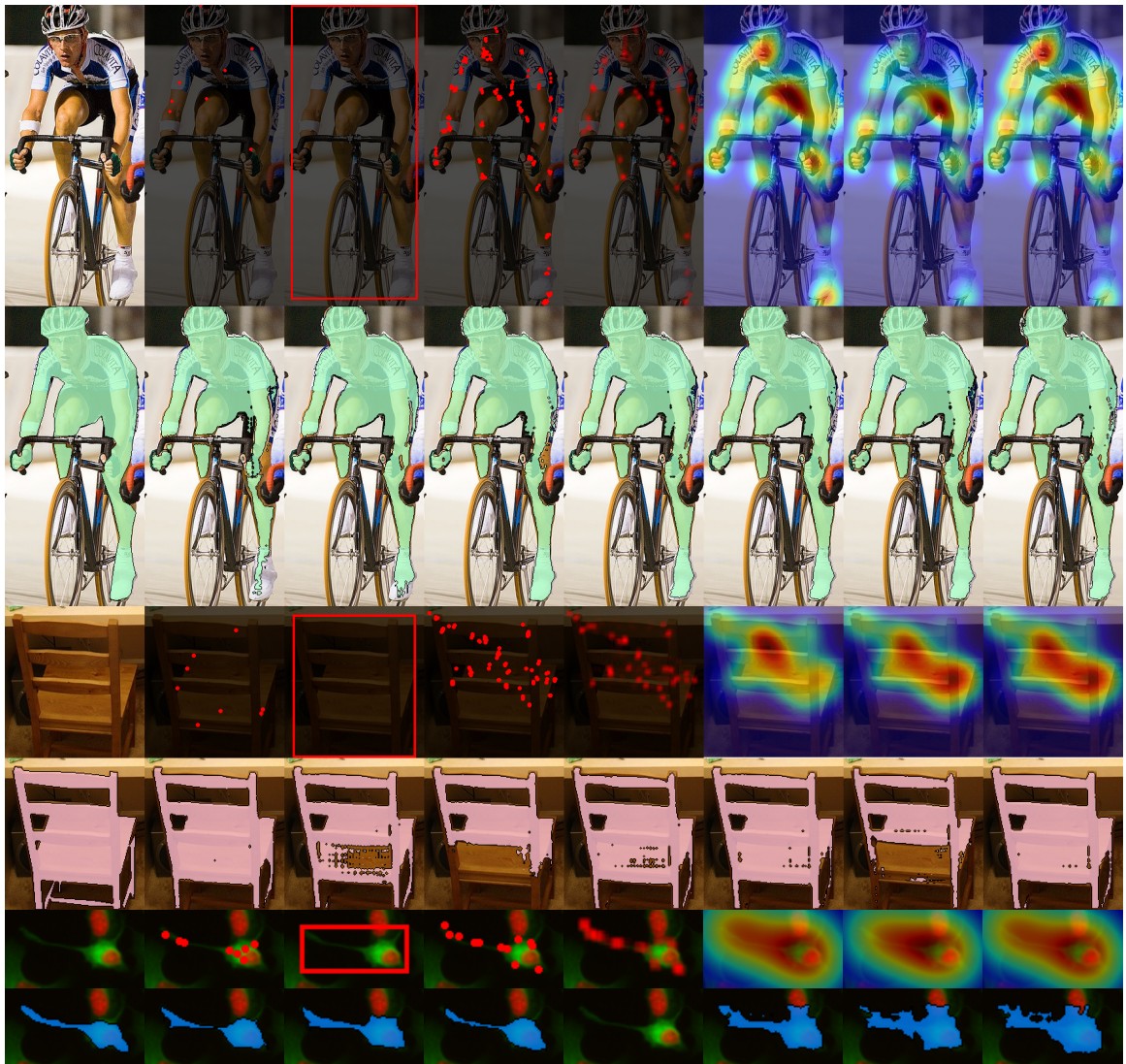

Figure 7: Comparison of the different prompt types and their corresponding segmentation results. **First Column:** Ground Truth Mask, **Second Column:** 8 Random Points, **Third Column:** Boundingbox, **Fourth Column:** Fixation Points, **Fifth Column:** Blurred Gaze, **Sixth Column:** Heatmap Gaze, **Seventh Column:** Heatmap Fixations, **Eighth Column:** Heatmap Fixations + Duration.

A qualitative evaluation of our method can be found in Figure 7. It shows mask predictions for the prompt inputs we evaluated. There are no significant differences for the cyclist. The point-based and the fixation method seem to generate the best masks. For the

chair example, larger qualitative differences can be seen. Especially for the fixation-based method the lower area of the chair which was not inspected by the human annotator, is also omitted in the segmentation mask. Interestingly, this is not the case for the "Blurred Gaze", "Heatmap Gaze" and "Heatmap Fixation + Duration" methods. The cell example from the Cellpose 500 dataset shows that the point-based method, together with the fixation method, generates the segmentation mask most similar to the ground truth. All three heatmap variants show large deviations and inaccuracies, which are probably due to the fact that the selected size of the Gaussian kernel is too large for smaller objects. Overall, the results show qualitatively that the masks, especially those generated with fixation-based prompts, are of equal or similar quality to the existing prompt methods of original SAM. In some cases, the accuracy of gaze-based generated masks might even seem more accurate, even if the IOU and F1 scores are worse. Additional qualitative results are shown in Appendix Figure A.5.

## 4. Conclusion

In this paper we propose the usage of eye tracking data as a new prompt type for the recently published Segment Anything model to quickly and accurately generate instance segmentation masks. For this purpose, we process the raw gaze points and generate fixation points, blurred raw gaze point maps and three heatmap variants based on the raw gaze points, fixation points and the fixations combined with their duration. We finetune and evaluate the sparse or dense encoder and the mask decoder of the original SAM model for each of the five variants. We are able to show that our method is faster in terms of annotation speed and achieves comparable segmentation performance. Since we record the eye tracking data passively and without a specific task, we believe that this method has great potential to help annotating datasets where expert knowledge is required and images need to be inspected by experts anyway, e.g. in the context of medical data. In order to evaluate this in more detail, we plan to further develop our method in different directions and make use of the temporal information contained within the raw eye tracking data to improve segmentation performance of neural networks as well as allow interactive annotations with different visual feedback.

**Data availability:** The gaze data for the PascalVOC2012 validation dataset are available at https://doi.org/10.17879/08958601057.

## Acknowledgments

JK, JG and BR would like to thank the Deutsche Forschungsgemeinschaft (DFG) - CRU326. DB, FK and BR would also like to thank the Deutsche Forschungsgemeinschaft (DFG) – CRC 1450 – 431460824. The calculations for this work were performed on the computer cluster PALMA II of the University of Münster. The authors would like to thank Nadine Bauer and Raghu Erapaneedi for generous sharing of data and constructive discussions and Sebastian Thiele and Leon Pielage for their feedback on the manuscript.

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

# Appendix A. Appendix

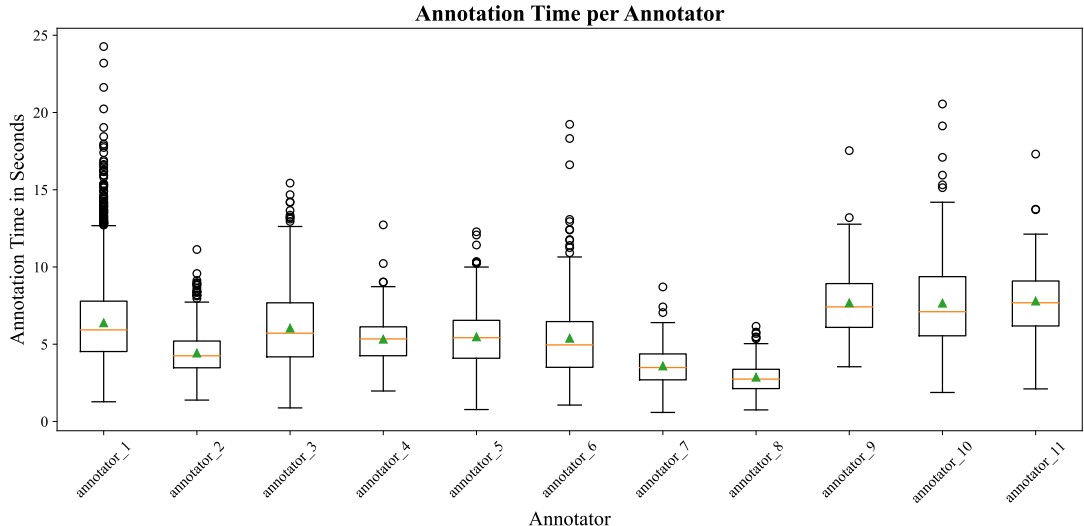

Figure A.1: Boxplot of the annotation time per human annotator for all datasets aggregated. The green triangles visualize the mean values.

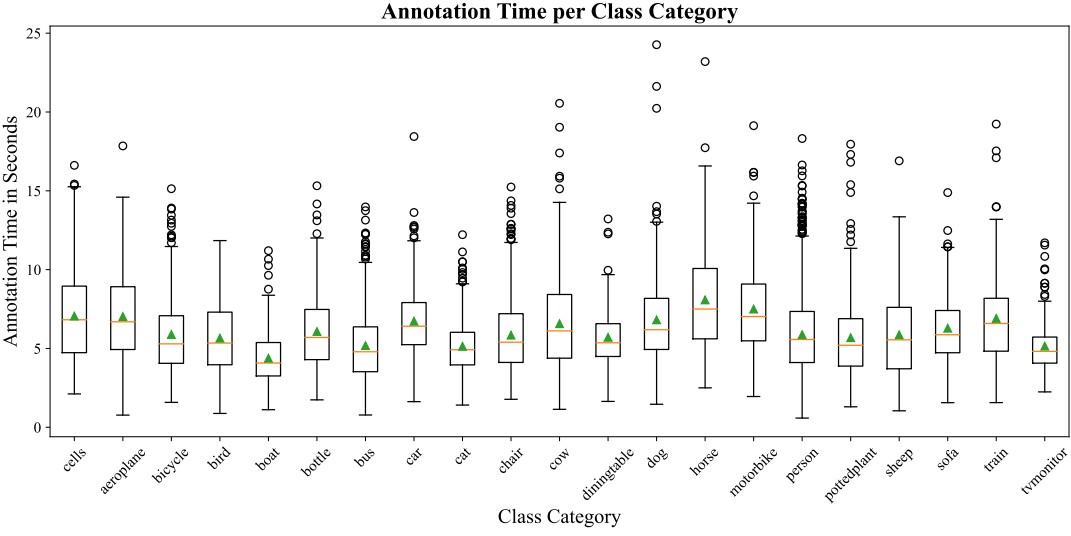

Figure A.2: Boxplot of the annotation time per class category for all datasets aggregated. The green triangles visualize the mean values.

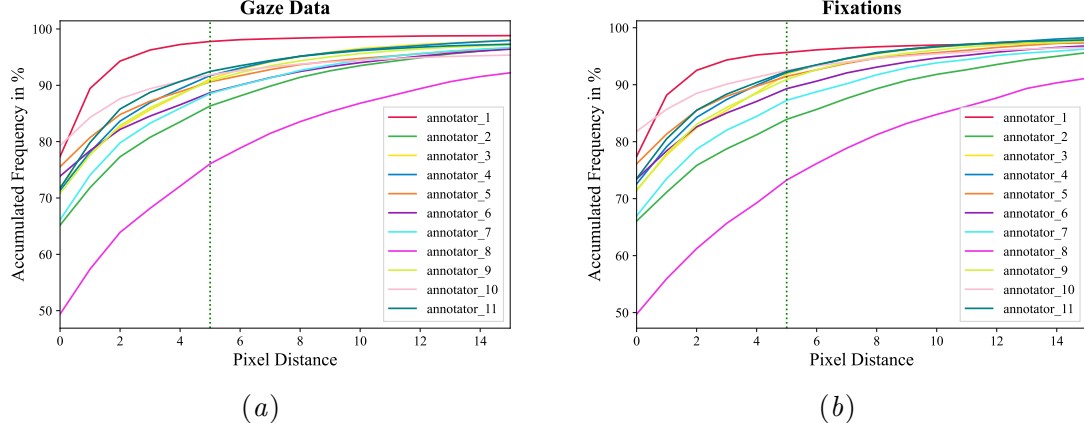

Figure A.3: (a) Raw Gaze Data and (b) Fixations grouped by their distance to the ground truth mask of all datasets grouped per human annotator.

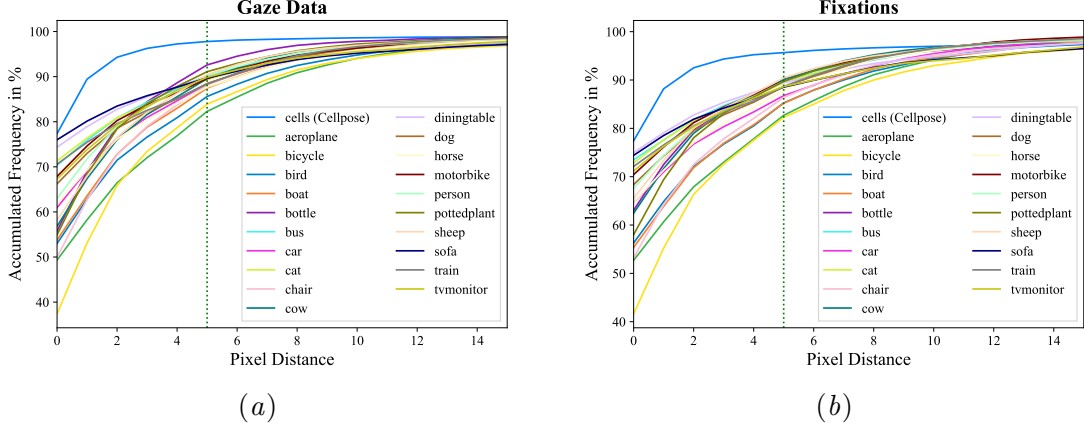

Figure A.4: (a) Raw Gaze Data and (b) Fixations grouped by their distance to the ground truth mask of all datasets grouped per class category.

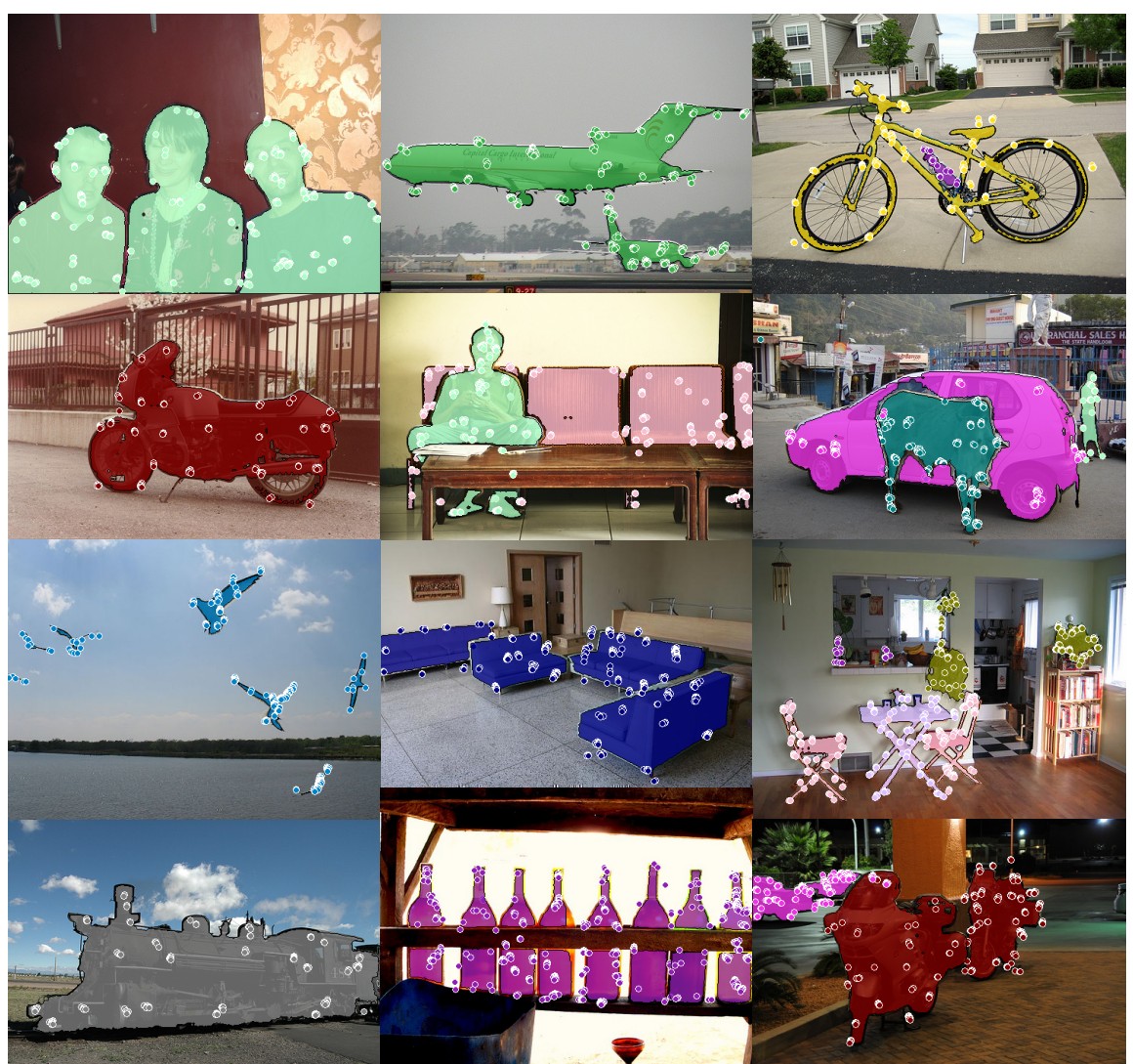

Figure A.5: Additional example results for fixation-based prompt segmentation.

