# OpenReview forum: "SAM meets Gaze: Passive Eye Tracking for Prompt-based Instance Segmentation"
_NeurIPS.cc/2023/Workshop/Gaze_Meets_ML — Gaze Meets ML 2023 Poster_

### Official Review · Reviewer_bJjt · 2023-10-22
**A basic method that accelerates the annotation by 2 secs with the utilization of gaze data as prompt in SAM for the cost of a 3% decline in accuracy**

**Rating:** 5
**Confidence:** 4

**Review:**

In the paper, authors propose utilization of gaze data, which obtained via passive eye tracking, as prompt input for segment anything model (SAM). They integrated a passive screen-based eye tracking system that allows annotators provide gaze input as prompt for segmenting a specific object instance.

The study aims to fine-tune the prompt encoders and mask decoder of the original SAM model for different gaze-based prompts. Two variants of gaze inputs are investigated: mask-based and point-based prompts. Mask-based prompts use four different types of input masks constructed directly from the recorded gaze. Point-based prompts use calculated fixation point coordinates as a sparse input, but this results in high memory consumption. All experiments use the publicly available pretrained ViT/H variant of SAM as a baseline. To reduce computational cost and training time, the heavyweight image encoder is frozen, training only lightweight downstream modules. The study uses a non-interactive training procedure and compares gaze-based prompts to finetuned versions of SAM using bounding boxes or sampled foreground points as input.

According to the quantitative results of segmentation accuracy (IOU) presented in Tables 1 and 2, the best result among the proposed methods belongs to the method using fixation points and is 2.94% and 2.81% behind the method using points, respectively, which makes the proposed method worse than comparable. Other proposed prompt mechanisms (i.e., blurred gaze, HM gaze, HM fixations and combinations of such) could perform way worse than the one utilizing fixations.

Although quantitative measurements are provided in the paper in terms of seconds, a detailed speed up analysis was not provided in the experimental results section. According to the information provided in the related work, we understand that the proposed method 2 seconds faster than the study of Papadopoulos et al. (2017b). Therefore, it can be inferred that the proposed method speeds up the annotation process by 2 seconds at the cost of a 3% decrease in segmentation accuracy.

The paper is well written and the quality of the visuals is of a high standard. I would also like to see a more detailed description of the annotation process. It is mentioned that there are no specific instructions or time limits, which might make sense for an annotation scenario where each image has a single object class. But I would like to see how a multi-class annotation scenario could be handled with the proposed method.
Some minor issues:
- In the caption of the Figure 7 "third column" was skipped.

In conclusion, considering the incremental novelty of the method (there are many other methods utilizing gaze and saliency) and the quantitative results that the paper presents make it around the threshold.

---

### Official Review · Reviewer_wgAY · 2023-10-23
**Thorough evaluation of different methods to combine eye-tracking data with the recently released Segment Anything (SAM) model**

**Rating:** 8
**Confidence:** 3

**Review:**

The paper proposes the use of data collected by an eye tracker with the Segment Anything foundational model (SAM) and studies different types of data inputs from the eye tracker, finding what works best. It also finetunes the SAM model to use each type of data as input to reach better segmentation scores.

Strengths
- Clarifies how to use eye tracker with SAM model: the paper tests several methods for inputting eye-tracking data to the SAM model: Spatial heatmaps vs Structured image coordinate sequence, Full gaze data vs Only fixations, Using the duration of fixations vs Not using it, Blurring data vs Not. It also compares results with additional finetuning of the SAM model. In a discussion comparing the use of SAM from this paper vs Training your own segmentation model from a reference, it shows some advantages of each of the approaches.
- Stratification of results: the paper shows that the time of segmentation increases with the average pixel area and with the average convexity of the segmented object. Experiments are run for 11 users, allowing to understand what kinds of variability is expected between users when running similar segmentations.

Weaknesses
- IRB or equivalent: the paper does not give enough details from the data collection sessions to indicate that it should be IRB exempt. It also does not specify the involvement of an IRB. The correct use of data collected from human subjects should be checked before the acceptance of the paper.
- Worse results with eye tracker: despite the claim that the eye tracker segmentations stay comparable to the mouse segmentations, there are differences of 1.5 percentual points to 2.5 percentual point in the different tests and scores run.
- Timing information not fully usable: The segmentation tool studied, different from usual interactive segmentation tools in this paper does not show the resulting segmentation of the model with current points while users can still add more points. Because of this, users cannot be sure of when to finish collecting gaze, and the timing data collected might be different than from the case where instant feedback is implemented. I understand that, since the paper was comparing different ways of using the gaze data for segmentation, having instant feedback would multiply the need of collected data by up to 8 times. However, I do believe the timing information is less usable in that case. I also believe that Annotator_8 (low annotation times, low gaze accuracy) might have done their data collection differently if feedback was provided to the users. Another problem with the analysis of the time data is that it is compared to data another paper from 2017, which might have a different data collection protocol, and a different final result than the baseline used to compare segmentation results, which had 8 points randomly selected from the ground truth segmentation mask.
- Unclear impact of calibration: it is not clear what impact calibration of the eye tracker has on the results. How much time does it take? What is the angular precision and inter/intra user variation? How often does it have to be repeated?

I can’t verify the IRB worry as a reviewer without a discussion board with the authors. The paper does extensive analysis of the proposed method and give useful insights about how to use the eye tracker model for interactive segmentation, which I believe can lead to useful discussion. The collection-time-related findings, which represent a good part of the paper, represent data collected in a very restricted scenario and might not be valid for more general scenarios.

---

### Official Review · Reviewer_iGTj · 2023-10-24
**Relevant merger of 2 significant arenas to enhance image segmentation and object detection**

**Rating:** 8
**Confidence:** 2

**Review:**

Quality: The paper was well-written and detailed. The corresponding research, datasets, training methods, results and evaluation techniques are described in good detail.

Clarity: For the most part, the paper was easy to follow and fairly easy to understand. The inclusion of Training approaches helped with the same.

Originality: The paper was transparent in acknowledging that the combination of gaze with supervised learning and image segmentation was a well researched idea and limits itself to the merger of SAM and gaze. That said, a paper called "GazeSAM: What You See is What You Segment" was published earlier this year - and it seems like that paper was published before the paper in consideration. GazeSAM focused on the medical domain, but this particular paper also discusses the relevant of SAM + gaze in that same domain. Therefore, I am a bit surprised on why the findings of GazeSAM were not discussed.

Significance: Tools and applications developed from combining SAM with gaze has the potential for a very obvious positive implication in the medical domain - something radiologists, surgeons, and other doctors can use with more confidence, as against the pure image segmentation models. That said, these benefits can translate into other domains, such as engineering, mining etc.

---

### Meta-Review · Area_Chair_9HEH · 2023-10-26

**Recommendation:** Accept (Poster)
**Confidence:** 4

**Metareview:**

Thank you for a well-written paper providing useful analysis of different ways of integrating expert gaze with SAM, including fine-tuning of masked-based and point-based gaze prompts. The reviewers feel that the paper can be strengthened with better documentation of the data collection process. Given that there is incremental novelty of the proposed method (as there are many other methods utilizing gaze and saliency), the meta reviewer would recommend accept for poster presentation on the basis of the thorough quantitative methods comparison results that the paper presented.

---

### Decision · Program_Chairs · 2023-10-26

Accept (Poster)